# Rotavirus Particle Disassembly and Assembly In Vivo and In Vitro

**DOI:** 10.3390/v15081750

**Published:** 2023-08-16

**Authors:** Dunia Asensio-Cob, Javier M. Rodríguez, Daniel Luque

**Affiliations:** 1Department of Molecular Medicine, Peter Gilgan Centre for Research and Learning, The Hospital for Sick Children, 686 Bay Street, Toronto, ON M5G0A4, Canada; dunia.asensiocob@sickkids.ca; 2Department of Structure of Macromolecules, Centro Nacional de Biotecnología/CSIC, Cantoblanco, 28049 Madrid, Spain; 3Electron Microscopy Unit UCCT/ISCIII, 28220 Majadahonda, Spain; 4School of Biomedical Sciences, The University of New South Wales, Sydney, NSW 2052, Australia; 5Electron Microscope Unit, Mark Wainwright Analytical Centre, The University of New South Wales, Sydney, NSW 2052, Australia

**Keywords:** dsRNA virus, reovirus, rotavirus, virus structure, virus assembly

## Abstract

Rotaviruses (RVs) are non-enveloped multilayered dsRNA viruses that are major etiologic agents of diarrheal disease in humans and in the young in a large number of animal species. The viral particle is composed of three different protein layers that enclose the segmented dsRNA genome and the transcriptional complexes. Each layer defines a unique subparticle that is associated with a different phase of the replication cycle. Thus, while single- and double-layered particles are associated with the intracellular processes of selective packaging, genome replication, and transcription, the viral machinery necessary for entry is located in the third layer. This modular nature of its particle allows rotaviruses to control its replication cycle by the disassembly and assembly of its structural proteins. In this review, we examine the significant advances in structural, molecular, and cellular RV biology that have contributed during the last few years to illuminating the intricate details of the RV particle disassembly and assembly processes.

## 1. Introduction

During their replication cycles, double-stranded RNA (dsRNA) viruses face several challenges inherent to the nature of their genomes. Since host cell enzymes cannot recognize dsRNA as a template for transcription, the virus must incorporate a transcription machinery that can synthesize the necessary mRNAs to initiate the viral cycle. In addition, dsRNA triggers an innate cell-based antiviral response, including interferon synthesis and apoptosis [1], which the virus must bypass to control the host response [2,3]. Most dsRNA viruses have evolved a common solution to these problems: they build a stable protein cage in the host cytoplasm that isolates the viral dsRNA molecules, preventing the cellular antiviral response. This cage, also known as the viral core, contains the enzymes necessary for the transcription and replication of the genome, which are carried out without disassembling the core particle. The core has an ordered architecture consisting of an icosahedral T = 1 shell formed by 60 asymmetric dimers, a 120-subunit capsid present in most dsRNA viral families [4,5]. While most of these viruses have a single protein shell and no extracellular cycle, the viruses belonging to the order *Reovirales* and to the genus *Cystovirus* have concentric protein layers surrounding the core, responsible for host cell recognition, entry, and other processes [6,7].

The members of the order *Reovirales* have non-enveloped particles and present a replication cycle that is regulated through the fine-tuned disassembly and assembly of the different proteinaceous icosahedral layers [8,9]. Due to their clinical relevance, rotaviruses (RVs), together with mammalian reovirus [10] and bluetongue virus [11], are the model systems for this order and their replication cycles have been extensively studied [12].

RV is the leading cause of severe gastroenteritis with dehydration in children under 5 years of age and causes ~590 million infections per year in all age groups. The introduction of RV vaccines has led to their inclusion in national immunization programs in >100 countries. Despite this progress, RV continues to be the cause of approximately 235,331 deaths (with a 95% confidence interval of 110,221–415,457), with the majority of fatalities occurring in developing countries [13]. In addition, RV represents a significant economic burden to health systems in developed countries, with an estimated 5.5 million infections and 1.6 million hospitalizations in children under five years of age in 2016 [13,14,15,16].

The genome of RV consists of 11 dsRNA molecules, with a total length of approximately 18,500 base pairs. These RNA segments encode six structural proteins (VP1, VP2, VP3, VP4, VP6, and VP7) and six non-structural proteins (NSP1 to NSP6). Each RNA segment is monocistronic, except for segment 11, which, in certain strains, contains two overlapping open reading frames (ORFs) encoding NSP5 and NSP6 [12,17]. The infectious virion is a non-enveloped, icosahedral, triple-layered particle (TLP), approximately 100 nm in diameter (Figure 1A), that resembles a wheel when visualized by electron microscopy [18,19,20]. In the past three decades, various structural, molecular, and cellular biology studies have revealed not only the structure of the RV virion but also how its multiple layers are disassembled and assembled during infection to perform various functions. In this review, we examine the structure of the virion and its subviral particles and review studies supporting the current models for the disassembly and assembly of these particles.

## 2. Rotavirus Virion Structure

The RV infectious particle is built by three concentric proteinaceous icosahedral layers that surround the viral genome and its replication/transcription machinery [20,21,24]. The inner core of rotavirus consists of a single-layered particle (SLP) composed of a T = 1 capsid formed by 60 asymmetric dimers of VP2 protein (102 kDa), enclosing eleven dsRNA genomic segments and associated with the RNA-dependent RNA polymerase VP1 (125 kDa) and the RNA capping enzyme VP3 (88 kDa) at the pentameric symmetry positions [20,21,24,25]. The SLP is surrounded by a thick T = 13 layer formed by 260 pear-shaped VP6 trimers (45 kDa) to form the so-called double-layered particle (DLP) [20,22,23,26]. The outer layer of the RV triple-layered particle (TLP) is formed by 60 trimeric VP4 spikes and 260 Ca^2+^-stabilized trimers of the VP7 glycoprotein [20,27,28]. Each of these protein shells has different chemical and biophysical properties that account for their different functions during the virus cycle [29]. The resolution of the in virio atomic structure of VP1, VP2, VP4, VP6, and VP7 (Figure 1C) has allowed us to understand the molecular interactions of these proteins in the viral particle. 

The inner T = 1 spherical shell, which is approximately 55 nm in diameter, consists of two comma-like-shaped VP2 subunits forming each asymmetric unit (Figure 1C). A star-like complex is formed by five copies of the VP2-A conformer around the icosahedral five-fold axis, while the VP2-B conformers are located in the gaps between the points of the star [20,23] (Figure 1B). The VP2 subunits form a continuous shell, with pores located at the five-fold axes. These channels are relatively small and have positively charged residues on their outer faces [30]. VP1 is situated on the inner surface of the pentameric positions (Figure 1A and Figure 2A), through contacts with the VP2 N termini [21,24]. This VP1–VP2 interaction not only stabilizes the RdRp on the inner shell surface but also plays an essential role in activating the polymerase activity [21,30].

The rotavirus RdRp, located with an offset from the five-fold symmetry axis, interacts with several of the surrounding VP2 molecules via complementary surface interactions and through their flexible N-termini. Thus, although the first 70 N-terminal amino acids of VP2 remain unresolved, three long N-terminal VP2 extensions that form tentacle-like interactions with VP1 (Figure 2A) have been characterized. Deletion of these N-terminal extensions prevents VP1’s incorporation into recombinant virus-like particles but does not hinder capsid–shell assembly [31]. The process of capping the nascent transcripts synthesized from the genomic dsRNA segments by the RV polymerase VP1 is performed by VP3, which in vitro is able to form a stable tetrameric assembly [32]. Each subunit of VP3 has a modular domain organization that integrates the five distinct enzymatic steps required for the capping of the transcripts in a unique way. However, in virio transcribing VP1 structures [21] suggest that nascent transcripts pass directly from the polymerase catalytic site through the five-fold pore to the capsid exterior. The process by which VP3 adds the 5′ cap to the transcript remains an unresolved puzzle. Different mechanisms, such as the re-entry of the transcript into the capsid interior or a re-threading mechanism, have been proposed [24].

**Figure 2 viruses-15-01750-f002:**
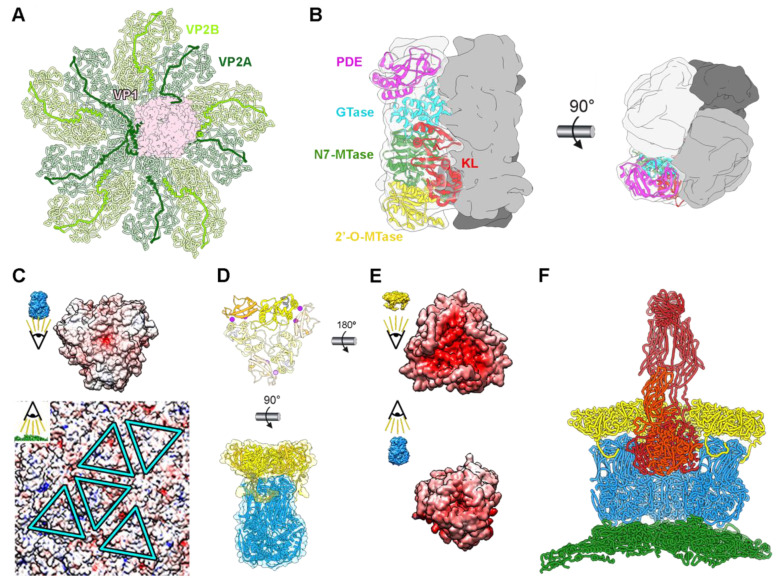
Rotavirus viral protein interactions. (**A**) In virio interactions of VP1 RdRp (pink) and VP2 decamer (green) (PDB 6OGY). The N-termini of VP2A (dark green) and VP2B (light green) subunits are highlighted. (**B**) Structure of the VP3 tetramer (PDB 6O6B). One subunit is shown with transparency, with its domains represented in separate colors: kinase-like (KL) domain (red), guanine-N7 methyltransferase (N7-MTase) domain (green), 2′-O-methyltransferase (2′-O-MTase) domain (yellow), guanylyltransferase (GTase) domain (cyan), and phosphodiesterase (PDE) domain (magenta). Front (left) and top (right) views of the complex are shown [32]. (**C**) Electrostatic potentials of the inner surface of a VP6 trimer (top) and of the outer face of the VP2 shell (bottom). The positions for the interactions of the five quasi-equivalent trimers on the VP2 surface are marked with blue. (**D**) Atomic structure of the VP7 trimer (PDB 3GZT) with one subunit highlighted (top). Two calcium ions (magenta) are bound at the interface between VP7 subunits. Interaction between VP6 and VP7 trimers (bottom). (**E**) Electrostatic potential of the inner surface of a VP7 trimer (top) and of the outer surface of a VP6 trimer (bottom). (**F**) VP4 interactions. Side view of the pentamer-contacting hexamers where the spike is located. VP2 is represented in green, VP6 in blue, VP7 in yellow, and VP4 in red. Only the back VP6 and VP7 trimeric capsomers are shown.

The SLP is enclosed by a T = 13 capsid, approximately 15 nm thick, composed of 260 pear-shaped VP6 trimers. These trimers adopt five distinct conformations, forming the 70 nm DLP (Figure 1A) [22,23]. Unlike the smooth surface of the core, the DLP exhibits an uneven surface with depressions located at the centers of pentamers and hexamers, providing access to the VP2 surface. This DLP serves as the transcriptional machinery for RV, producing capped, non-polyadenylated, positive single-stranded (ss) RNA that can effectively initiate an infection upon release into the host cell cytoplasm [33]. The assembly of the T = 13 layer of VP6 on top of the T = 1 layer that form the 60 VP2 dimers on the SLP represents a notable example of symmetry mismatch, a characteristic preserved in most reoviruses. This asymmetry has been proposed to regulate the polymerase activity [23,24]. VP6 trimers settle onto the hydrophobic outer surface of VP2, occupying five distinct positions (Figure 2C, triangles). This positioning is facilitated by hydrophobic interactions involving the inward-projecting loop 64-72 of VP6, which encounters the outer surface of the SLP. Intertrimeric contacts between VP6 molecules occur through their pedestal domains, forming local two-fold contacts. While the VP2–VP6 and intertrimeric VP6–VP6 contacts are relatively modest, they play crucial roles in both assembly and transcription processes [34].

DLP are non-infective when added to cells [35,36], but they can cause infection when transfected into the cytoplasm [33]. The reason for their non-infectivity lies in the inability to recognize, bind to, and penetrate the host target cell, capabilities attributed to the outer layer of the TLP comprising VP4 and VP7. VP7, a glycoprotein, forms 260 Ca^2+^-stabilized trimers that embrace each VP6 trimer of the DLP through its N-terminal arm (Figure 2E) [27,28,37]. The VP7 trimer’s bottom surface has minimal contact with the VP6 trimer’s apex, the interaction between both proteins being primarily facilitated by the VP7 N-terminal arm embracing the underlying VP6 trimer. These arms also interact with adjacent VP7 trimers, creating a cooperative lattice that reinforces the RV outer shell. It has been suggested that calcium ions do not only stabilize the VP7 trimers but also serve as a bridge between the VP7 inner and VP6 outer surfaces, being sandwiched between them to allow their assembly [29]. In this scenario, the depletion of calcium would destabilize the VP7 intertrimeric interactions, leading to the rapid disassembly of the shell through the disruption of VP7–VP6 electrostatic interactions (Figure 2D).

Finally, the viral spike consists of three copies of VP4, which are proteolytically processed by trypsin-like proteases from the intestinal lumen to produce VP5* (60 kDa) and VP8* (45 kDa), resulting in a fully infectious virion [20,38,39,40]. Each viral spike is formed by three VP4 units and is anchored to the depressions in the center of the pentamer-contacting hexamers (Figure 2F). The spike is an extreme example of structural polymorphism with trimeric, dimeric, and asymmetric elements (Figure 1C and Figure 2F). The C-terminal domains of the three VP5* subunits interact, forming a trimeric foot that sits between the VP6 and VP7 layers. While the connection between the spike foot and VP7 is relatively weak (mediated by the VP7 N-terminal arms), the assembly of VP7 trimeric caps onto VP6 secures the spikes by narrowing the diameter of the cavity above the spike base. The region protruding from the VP7 layer lacks the local trimeric symmetry observed in the foot. The beta-barrel of one subunit forms the spike stalk, while the beta-barrels from the other two subunits create a dimeric spike body that extends outward from the particle surface, capped by two lectin domains.

## 3. Particle Disassembly during the Viral Cycle

Particle disassembly and assembly during the viral cycle are closely coordinated with changes in the cell environment. Disassembly begins and progresses in response to cellular signals that trigger conformational changes in the viral particle. A precise sequence of cellular cues and conformational responses by the viral machinery allows the particle to proceed from the initial receptor interaction in the cellular membrane to the precise endocytic compartment where membrane rupture is possible [41]. Likewise, particularly in segmented viruses of the order *Reovirales*, assembly progresses in stages where the particle structure and composition are coordinated with changes in the cellular environment produced by the progression of the virus cycle [42].

In most dsRNA viruses, entry of the infecting virion results in the partial disassembly and release to the cytoplasm of a transcriptionally active core particle that does not further disassemble [1]. These characteristic ribonucleoprotein complexes produce ss(+)RNAs that function as mRNA for the translation of viral proteins or as precursors of the viral genome, while simultaneously protecting the dsRNA genome from cellular surveillance mechanisms [43].

In rotavirus, partial disassembly of the TLP occurs during viral entry and is associated with membrane disruption [44]. Entry ends with the liberation of the transcriptionally active DLP particle into the cytoplasm, effectively initiating infection [33]. The machinery necessary for entry constitutes the outer layer of the TLP [9,45]: the VP7 capsid, which transforms the transcriptionally active and soft DLP into mechanically strong [29], transcriptionally inactive, TLP; and VP4 spikes, responsible for target acquisition [9,46,47] and entry route selection [48], whose conformational changes are the main drivers of membrane penetration [49].

Arguably, rotavirus disassembly (Figure 3) begins with the activation of the TLP particle by the action of trypsin-like proteases in the digestive tract of infected animals [38,39,50]. Cleavage occurs in the exposed loops of the three VP4 chains and, as a consequence of the different structural conformations adopted by the chains, after digestion, VP4-A and -B maintain the lectin-like VP8* domain noncovalently bound atop VP5*, while the lectin domain of VP4-C, which has not been identified in any of the structures determined so far, is presumed to be released from the particle [20,51,52]. This activation produces minimal but essential changes in the structure of the spikes [52,53], making the TLP competent for productive entry [54].

Rotavirus initially engages the cell through the distal VP8* domains of VP4 chains A and B, which, depending on the genotype, have been reported to interact with sialic acid, gangliosides, histoblood group antigens, or mucin cores [9,46,55]. This interaction appears promiscuous since only a few of the cell lines to which rotavirus binds are efficiently infected [56]. Nonetheless, this initial interaction influences RV tissue tropism, host range restriction, and interspecies transmission [46,57,58]. Several post-attachment interactions of VP5* and VP7 have been described using cell lines, polarized intestinal cells, or human intestinal enteroids, and involving gangliosides, different integrins, the heat shock cognate protein hsc70, occludin, and the tight junction protein JAM-A. However, blockage of these interactions produces a moderate reduction in infection titers, suggesting a high redundancy/plasticity of virus entry interactions or the presence of yet unknown key receptor proteins [55,59].

Virus internalization occurs in most rotavirus strains by clathrin-mediated endocytosis dependent on cholesterol and dynamin [48,55,59,60,61]. The RRV simian strain is exceptional in that it follows a different endocytic pathway, independent of clathrin and caveolin, but requires cholesterol and dynamin [61]. The endocytic route employed seems to be determined, due to an unknown mechanism, by the nature of the spike protein VP4; thus, a single amino acid mutation on VP4 shifts the entry of RRV to a clathrin-dependent process [48]. Regardless of the endocytic pathway followed, the particles reach an early endosome [62,63,64] and progress to maturing endosomes, from which some strains (termed early-penetration strains) can be released to the cell cytoplasm and initiate transcription. Late-penetration strains must reach late endosomes before they are released to initiate a productive infection [55,62,63,65].

Independently of the entry route and the final endosomal compartment that it reaches, the incoming particle must penetrate the cell membrane to initiate infection. The current model for this last step of virus entry has been established using the RRV strain in BCS-1 cells [44,45,49,66,67,68]. In this system, entry does not depend on clathrin or dynamin and is believed to rely solely on the viral entry machinery [44]. The determination of the near-atomic structure of a new reversed conformation of the VP4 spike has improved our understanding of the molecular mechanisms involved [49]. Initial binding to the cell membrane occurs through the interaction of the distal lectin domains of the two VP8* molecules of chains A and B with a sialylated ganglioside. After attachment, progressive interactions of the adjacent lectin domains with the cell membrane initiate the invagination of the particle, followed by engulfment in a loose-fitting vesicle [44].

**Figure 3 viruses-15-01750-f003:**
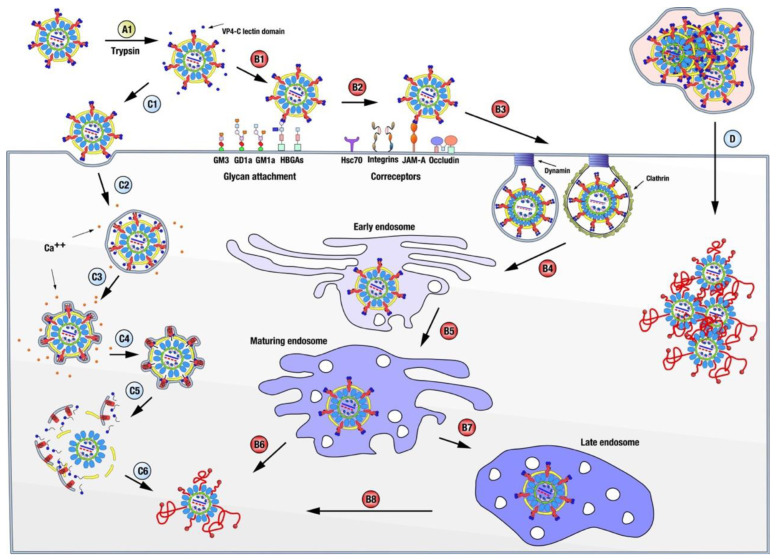
Rotavirus disassembly. Rotavirus disassembly begins with the cleavage of the outer capsid protein VP4 by trypsin-like proteases (A1), which release the lectin domains of the VP4-C chains from the virus. Rotavirus internalization progresses sequentially through interactions with glycan attachment molecules (B1) and coreceptors (B2), followed by endocytosis, which, depending on the strain, could be clathrin-dependent or independent (B3). Incoming viral particles follow the classical endocytic pathway (B4), reaching maturing endosomes (B5). From here (B6), some strains (termed early-penetration strains) can be released into the cell cytoplasm and initiate transcription. In contrast, late-penetration strains must first reach the late endosome compartment (B7) before they can be released to initiate a productive infection (B8). Our current model for membrane penetration (C1–C6) has been established using the RRV strain in BCS-1 cells and relies primarily on viral components. Initial binding to the cell membrane occurs through the interaction of the distal lectin domains of the two VP8* molecules of chains A and B with a sialylated ganglioside. Following attachment, progressive interactions of the adjacent lectin domains with the cell membrane initiate the invagination of the particle (C1), which is then engulfed in a loose-fitting vesicle (C2). Spontaneous fluctuations lead to dissociation of the lectin domains, revealing the two hydrophobic loops beneath them and triggering their irreversible insertion into the membrane. This is followed by the reorganization of the ß-barrel domains, adopting a trimeric structure in which the hydrophobic loops of the three ß-barrel domains are inserted into the membrane (C3). The free energy released during this process is likely the driving force for the wrapping of the virus in a tight-fitting membrane (C4). Finally, the spikes transition to the reversed conformation, which occurs when the amino acids of the foot domain of each of the three subunits, previously located in the foot cavity, are pulled towards and inserted into the membrane. The insertion of multiple foot domains (C4) leads to perforation (C5) and the release of DLP into the cytoplasm, starting transcription (C6). A third mechanism of entry for rotavirus, vesicle-mediated en bloc transmission, has been recently discovered to play a significant role in transmission [69], but it remains to be characterized further (D).

During this process, spontaneous fluctuations lead to the dissociation of the two VP8* lectin domains, exposing the hydrophobic loops of the two VP5* beta-barrel domains that lie beneath and allowing their irreversible insertion into the membrane. This is followed by the reorganization of the beta-barrel domains, adopting a trimeric structure in which the hydrophobic loops of the three beta-barrel domains are now inserted into the membrane. The free energy released during this process is likely to be the driving force for wrapping in a tight-fitting membrane [49] The presence of the three proposed stages of this model, the particle bound to the membrane, loosely enveloped, and wrapped, was confirmed by thin-section electron microscopy [44]. Finally, a long, three-strand, α-helical coiled coil is formed by polypeptide chain segments C-terminal to the beta-barrels, which, in the normal VP4 conformation, reside in the foot cavity. The formation of the coiled coil probably drives the pulling of the destabilized, partially unfolded remainder of the foot domain of VP4 through cavities in the trimeric beta-barrel structure, without disturbing the trimeric arrangement of the beta-barrels or its interactions with the VP7. The result is that, in this reversed conformation, the foot cavity beneath the VP7 layer is empty, and the ≈250 amino acids that constitute the foot domains of each of the three subunits, previously located in the foot cavity, are pulled towards and inserted into the membrane. Membrane destabilization is thought to occur through the insertion of multiple foot domains, leading to perforation and DLP release into the cytoplasm.

This model is supported by studies of cryo-electron tomography of RRV particles that infect BSC-1 cells. Analysis of tomograms revealed engulfed RRV particles that maintained two different distances from the outer layer of VP7 to the inner surface of the membrane, bridged by projections of VP4 [44,49]. Icosahedral averages of individual subtomograms obtained from nearly fully engulfed particles showed that, in loose-fitting vesicles, this distance is similar to that of the spikes on normal TLP, which is interpreted as the VP8* domains interacting with the attachment factors. The tight-fitting membranes are separated by a shorter distance, similar to the distance from the surface of the VP7 layer to the hydrophobic loops on the tip of the trimeric beta-barrel, which implies that the extruded foot domains are embedded in the membrane. Crucially, in these tomographic reconstructions, the volume occupied by the foot domains in the normal TLP appears to be empty, supporting the role of the reversed conformation during entry [49].

The flux of Ca^2+^ ions from the vesicular compartment surrounding the virion always precedes the onset of VP7 and VP4 dissociation by approximately 2 min (which leaves the particles together) [68]. Ca^2+^ leakage into the cytosol could follow any or both of the interactions of VP5* with the membrane involved in this model. A reduction in the Ca^2+^ concentration in the vesicle, which induces the depolymerization of VP7, is followed by the liberation of the DLP into the cytoplasm. Within 5 min of attachment, the particles are completely enveloped, being inaccessible to external agents, and within 3–5 additional min, the DLP contained in the wrapped TLP is released into the cytoplasm [44].

## 4. Particle Assembly during the Viral Cycle

Immediately after the DLP is released into the cytoplasm, positive-sense, capped, non-polyadenylated RNAs are extruded from the viral particle and initiate the synthesis of viral proteins (Figure 4). It is thought that virus assembly begins with the specific interaction of VP1, the viral polymerase, with conserved bases at the 3′ ends of ssRNA(+) that serve as precursors of the dsRNA segments, forming a pre-core complex in which VP1 is inactive [70,71,72,73]. These 11 different cytoplasmic complexes are specifically recruited to the viroplasms, which are cytoplasmic electron-dense inclusions where viral ssRNAs and viral proteins accumulate [74,75]. It is in the viroplasms that most of the events of rotavirus morphogenesis are compartmentalized. The generation of SLP, genome replication, DLP assembly, and secondary transcription occur in viroplasms [76]. Viroplasms are assembled by the interaction of NSP2 and NSP5 [77,78] and are dynamic structures regulated by phosphorylation events in these proteins [79,80,81]. Viroplasms are associated with lipid droplets [82,83,84,85], tubulin [86], and other cellular components [87,88].

Recently, viroplasms have been shown to act as biomolecular condensates [74,75,78,89,90] formed by the liquid–liquid phase separation (LLPS) of the proteins NSP5 and NSP2, where NSP5 acts as the main driver of LLPS (scaffold), of which NSP2 is a client protein [89]. Immediately after infection, viroplasms show liquid-like behavior that matures to solid-like condensates via the accumulation of other viral RNA and proteins that participate in these condensates as clients and by post-translational modifications, particularly the phosphorylation of NSP5. Studies using the superresolution microscopy of viroplasms have led to the description of viroplasms as highly organized structures where the different viral proteins are distributed as concentric layers enriched in a particular protein around a center formed by NSP5 [90]. Interestingly, multilayered behavior is a characteristic shared by many biomolecular condensates [91].

Particle assembly progresses inside the viroplasms by the assortment of the 11 ssRNA(+)s that will compose the genome of the mature particle [42,74,76,92,93,94]. This enigmatic process has recently been suggested to be driven by the interaction of NSP2 with the ssRNA(+), which alters its structure, revealing otherwise hidden complementary sequences capable of inter-segment base pairing [95]. It is believed that complexes containing the complete complement of segments and the polymerase machinery (VP1/VP3) nucleate the assembly of VP2 decamers around them, displace NSP2–NSP5 interactions with the ssRNA(+), and result in the assembly of the SLP particle. However, a recently published in situ analysis of rotavirus viroplasms by cryo-electron tomography shows the presence of genome-less SLPs, an unanticipated intermediate in current models of assembly [52].

The structure of genome-less SLPs displays a profound indentation at the five-fold symmetry axes due to the displacement of the VP2 dimers in these axes by 35º towards the center of the structure. A density compatible with VP1 polymerase appears on top of the indentations on each five-fold vertex. A similar genome-less SLP intermediate, but lacking the polymerase density at the five-folds, has been described for mammalian orthoreovirus [96]. Both are reminiscent of intermediate stages of assembly found in bacteriophages of the *Cystoviridae* family, suggesting that this indented pre-packaging stage is a conserved feature across the order *Reovirales* and the *Cystoviridae* family [52]. How this new particle fits into the current assembly model and how the VP1–VP3–ssRNA(+) complexes access its interior remain to be elucidated. Once the genome containing SLP is formed, the interaction of VP2 with VP1 activates polymerase activity and allows the replication of genetic material [30,71,97]. At the peripheries of viroplasms, where VP6 is concentrated [76,90], the assembly of the VP6 capsid transforms the SLPs into DLPs. Newly assembled DLPs are transcriptionally competent and initiate a secondary transcription that increases ssRNA(+) production [98,99,100].

Until now, the morphogenesis of rotavirus particles has been considered a purely cytoplasmatic process; however, the acquisition of the outer layer of the TLP occurs inside membranous cisterns that surround viroplasms by a series of processes that are not well understood. Recent results have shed light on the origin of these cisterns, which previously were thought to be of ER origin. In this new model [101,102], NSP4 and VP7 are initially located in the ER, where NSP4 interacts with the COPII transport system cargo binding protein Sec24 [103] and integrates, along with the protein VP7, into COPII vesicles that are released into the cytoplasm. Subsequent interaction with NSP4 inserts the autophagy marker protein LC3-II [104] into the vesicles and diverts them to the viroplasms’ periphery. Morphogenesis progresses by the budding of the DLP through the COPII-derived membranes. This process is initiated by binding of the C-terminal cytoplasmic domain of NSP4 to VP6 [105,106,107], which forms the outer coat of the DLP [42]. The incorporation of the spike protein VP4 is thought to occur from the cytosol during the budding process by interaction with NSP4 [108]. Progressive interactions of the outer DLP layer of VP6 with NSP4/VP7-containing membranes result in the budding of the DLP–VP4–NSP4–VP7 complex into the lumens of COPII-derived vesicles, in the form of transient membrane-enveloped particles (eDLP), which are a unique intermediate among the morphogenesis of dsRNA viruses. Recent data from the cryo-electron tomography of eDLPs have allowed an initial, low-resolution glimpse into its structure [52]. eDLPs appear as DLPs containing 60 trimeric VP4 spikes that attach the particles to the transient envelope. VP7 and NSP4 are not visible in these reconstructions. VP4 spikes in eDLPs show a broadly three-fold symmetric structure and are highly flexible. This premature conformation is reminiscent of the trimeric intermediate adopted by the spike during entry, where the foot domain is still in its cavity and the ß-barrel domains form a trimer with the hydrophobic loops inserted in the membrane. However, in the premature conformation, the VP7 layer is not present and VP4 is not digested; thus, the three lectin domains are each bound by two loops to their VP4 molecule. The transition from this trimeric premature conformation to the partially dimeric mature conformation observed in the TLP requires major rearrangements of the VP4 subunits, which are presumed to be responsible for the disruption of the transient envelope [52]. During this process, the polymerization of the VP7 layer onto the DLP particle locks the VP4 spikes in place [36,42,52].

The mechanism by which the fully assembled TLP, inside COPII-derived vesicles, moves to the outside of the cell has not been thoroughly characterized. Nevertheless, rotavirus appears to operate on more than one exit route. In the non-polarized kidney epithelial cells MA104, rotavirus is released by cell lysis [109], whereas in a polarized intestinal epithelial cell line, Caco-2, rotavirus is also actively secreted, before any cell lysis occurs, from the apical cell surface, trafficking by a novel vesicular transport that bypasses the Golgi apparatus and the lysosomes [110].

Until recently, viral entry, egress, and transmission were thought to occur essentially in all viral types through single, free virus particles. It is now clear that viruses can also be released from infected cells and transmitted as groups of particles protected inside extracellular vesicles, a process common to multiple viral types that has been called vesicle-mediated en bloc transmission [111]. Egress from the cell is non-lytic and applies to enveloped [112] or non-enveloped [113] viruses or even to infectious genomes [114]. The extracellular vesicles containing viruses can originate from different organelles, including autophagosomes [115], plasma membranes [69], secretory lysosomes [113], and multivesicular bodies [116]. The protection offered by the cloaking vesicle and the multiplicity of genomes that initiate infection on vesicle-mediated en bloc transmission have important effects on viral stability, replication fitness, the modulation of viral genetic diversity and evolution, and the viral response to immune recognition [111,117,118,119].

During rotavirus infection, vesicle-cloaked particles are non-lytically released from the cell [120] as large vesicles that originate at the plasma membrane [69]. They are also found in the stool of infected animals, where they are a non-negligible fraction of the total virus, significantly more infectious to animals than the equivalent number of free particles [69], which underscores the relevance of this transmission mechanism. Interestingly, vesicle-cloaked viruses released from H96 cells or found in the stool of infected animals show a processed VP4 [69], whereas the virus released from CaCo2 and MA104 cell lines have an intact VP4 [69,120]. Until now, the activation of VP4 was thought to occur in free particles by the action of intestinal trypsin-like proteases, but the presence of proteolyzed VP4 inside extracellular vesicles implies the existence of a new cell-specific activation mechanism. It is also noteworthy that vesicle-cloaked viruses released from MA104 cells, i.e., with an intact VP4, are infectious, raising the possibility of a different form of entry, independent of VP4, for vesicle-cloaked viruses [120].

## 5. In Vitro Disassembly and Assembly of RV Particles

Many viruses coordinate calcium ions in their particle structures to stabilize their monomers and the interface between their capsomers [121,122]. These calcium ions are essential to maintain structural integrity and in regulating assembly/disassembly processes [122]. We have seen previously that, in rotavirus, calcium ions stabilize the interaction between the VP7 trimers that constitute the external layer of the mature virion [28,123]. Additionally, calcium ions are proposed to play a role in mitigating the repulsion between the inner VP7 and outer VP6 electronegative surfaces, which are essential for TLP assembly [29]. In essence, TLP integrity is dependent on the calcium concentration, and, during rotavirus entry, the decrease in calcium concentration characteristic of endocytic vesicles is used by the incoming particle to facilitate VP7 disassembly, membrane penetration, and the release of the DLP into the cytoplasm [28,121,122,123].

This disassembly process has been emulated in vitro by means of chelating agents, such as ethylenediaminetetraacetic acid (EDTA) [40] or ethylene glycol-bis (beta-aminoethyl ether)-N,N,N′,N′-tetraacetic acid (EGTA) [124], which induce the dissociation of VP7 trimers by depleting the calcium ions that stabilize them (Figure 5A–C). These studies initially highlighted the crucial role of calcium ions in the regulation of the VP7 layer disassembly and assembly. Additionally, the purified DLP can be uncoated and converted into single-layer particles (SLPs) using chaotropic agents such as CaCl_2_ (Figure 5A,D) [125].

The mechanical properties of the different particle shells have been explored by atomic force microscopy using single indentation assays and mechanical fatigue experiments [29]. The strong VP7–VP7 and VP7–VP6 interactions provide high mechanical strength for protective purposes (Figure 5E, TLP). This resistance allows the TLP to overcome the severe extracellular conditions, including the stringent physicochemical conditions of the digestive apparatus. In contrast, the VP6–VP6 and VP6–VP2 interactions offer lower resistance, facilitating the required conformational dynamics for transcription. Fatigue assays in the presence of EDTA have allowed to live-image the rapid disassembly of the VP7 lattice when these ions are depleted (Figure 5E, TLP + EDTA). When mechanical fatigue is applied to DLP or partially disassembled TLP [29], the VP6 subunits are removed rapidly from the underlaying SLP (Figure 5E, DLP). This reflects the low mechanical stability of the trimeric VP6 layer [29] and its high flexibility, which allows the virus to adopt some level of deformation, necessary for the expression of its genome. Finally, the fatigue applied to the SLP after the disassembly of the VP6 layer indicated high instability in the VP2 layer (Figure 5E, SLP).

In vitro recoating has been used successfully in reovirus to study the assembly and cell entry mechanisms [126,127,128,129]. During recoating, the infectious cores, equivalent to rotavirus DLP, are incubated with the recombinant overexpressed components of the outer layer. This generates particles that are thousands times more infectious than cores but half as infectious as native virions [130,131]. Transcapsidation experiments showed that it is possible to recoat the rotavirus DLP with virion-derived outer capsid proteins to generate in vitro infectious particles [132]. However, the reaction is inefficient relative to the number of recoated particles. Nevertheless, DLP recoating with baculovirus expressed recombinant VP4 and VP7 proteins, produced at an acidic pH in the presence of calcium ions, generates highly infectious recoated rotavirus particles [36]. The recoating effectivity is strongly pH-dependent, with maximum efficiency at pH 5.2 and a lower effect at pH 7.2. These recoated particles are as infectious as authentic purified virions, and when recoated particles are incubated with EDTA, the infectivity is reduced to the level of DLP, accordingly with the drop of the infectivity in native TLP when they are depleted of calcium [35]. The low VP7 concentration needed for recoating assays suggests that VP7 binds DLP with high affinity. However, most of the particles show incomplete outer capsid recoating. This indicates a cooperativity binding process between VP7 trimers during assembly that creates patches of assembled VP7 in the particles but not fully coated or completely uncoated particles [29,36]. This was also evidenced during the dissociation of VP7 trimers in disassembly, or uncoating, assays [29,36]. VP7 binding does not depend on the prior assembly of VP4 in the DLP, but, for the full restitution of infectivity, VP4 must be added before VP7 [36]. VP4 spikes bind to DLP via relatively weak interactions; afterwards, VP7 trimers assemble around the particles and lock VP4 in its place.

Despite its relatively recent development, recoating has emerged as a powerful tool to study the role of the outer layer proteins since it allows the construction and study of mutants with lethal or very low fitness mutations, which are at present very difficult to generate. Furthermore, DLP recoating represents a key tool for the high-resolution cryo-EM analysis of different RV particle membrane penetration intermediates (Figure 6) and has been crucial in understanding how the refolding of the RV spike mediates membrane penetration [49].

## 6. In Vivo Assembly of RV Virus-like Particles

The structural proteins of numerous viruses self-assemble in vivo to form virus-like particles (VLPs), which possess a similar structure to the subparticles and particles generated during the natural viral cycle [133]. The expression of RV structural proteins through various heterologous expression systems has enabled the isolation of different VLPs from strains of group A rotavirus (RVA) and group C rotavirus (RVC), including VLPs with one layer (pseudo-SLP, vSLP), two layers (pseudo-DLP, vDLP), or three layers (pseudo-TLP, vTLP). VLP assembly not only requires the expression of various structural proteins but also their proper folding and assembly. The VLP assembly process in the different heterogeneous expression systems is a poorly characterized process that appears to occur spontaneously through the interaction of overexpressed structural proteins [134,135,136,137,138,139,140,141].

The expression of RVA VP2 allows the formation of self-assembled vSLP (Figure 7(A3)) [142]. These vSLP may also contain VP1 or VP1 and VP3 if co-expressed alongside VP2 [143]. Co-expression of VP2 and VP6 [144] leads to the assembly of vDLP (Figure 7(A4), where VP2 serves as the scaffold protein on which VP6 assembles [145]. The expression of VP6 alone does not produce icosahedral assemblies but results in the formation of spherical particles (Figure 7(A2)), 2D crystal matrices, or helical tubes (Figure 7B) as a function of the pH level [146,147,148]. Depending on the combination of co-expressed structural proteins, vDLP can be formed by VP2/6 (Figure 7(A4)), VP1/2/6, VP2/3/6, VP1/2/3/6, or VP2/4/6. The formation of vDLP (VP2/6) is crucial in generating vTLP, as VP7 and VP4 wrap around the VP6 layer within the vDLP. These vTLP can contain the RV spikes, VP2/6/4/7 (Figure 7(A6)), or be composed solely of structural proteins from the three concentric capsid layers VP2/6/7 (Figure 7(A5)) [134,143]. In addition to those of RVA, VLPs have only been generated for RVC strains, consisting of vDLPs (VP6/7, Figure 7(A7)) or vTLPs (VP2/6/7, Figure 7(A8)). It is also possible to generate hybrid vTLP assembled with structural proteins from different RV species, as has been done with RVA and RVC (A-VP2/C-VP6/C-VP7, A-VP2/C-VP6/A-VP7, and A-VP2/A-VP6/C-VP7) [149].

Several expression systems have been used to generate RV VLP, such as yeast (*Saccha-romyces cerevisiae*) [135], prokaryotic cells (*Escherichia coli* BL21 (DE3)) [140], transgenic plants (*Nicotiana tabacum*, *Nicotiana benthamiana*, *Lycopersicon esculentum*) [136,137,141], transformed *Drosophila melanogaster* cells [138], or mammalian cells (Vero 2-2 cells) [139]. However, most of the studies have used recombinant baculovirus (rBV) to express the heterologous proteins in insect cells. This system presents several characteristics that make it convenient for the expression of RV VLP [150,151,152,153,154], such as the baculovirus’ very late promoters that can promote high levels of expression, as well as the simultaneous co-expression of several proteins by means of bicistronic or multicistronic vectors.

**Figure 7 viruses-15-01750-f007:**
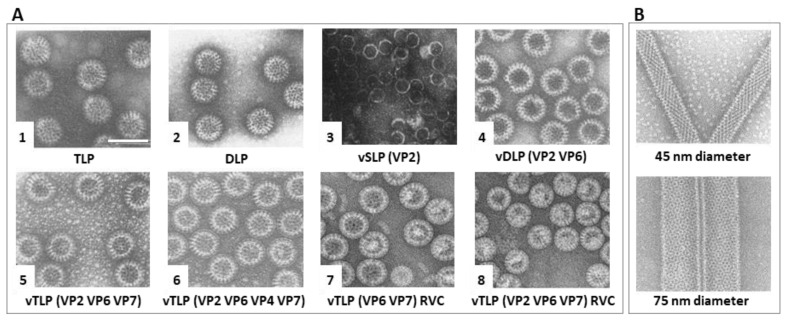
RV virus-like particle assembly in insect cells. EM fields of negatively stained (**A**) (**1**) native RVA TLP; (**2**) native RVA DLP; (**3**) RVA vSLP (VP2); (**4**) RVA vDLP (VP2/6); (**5**) RVA vDLP (VP2/6/4); (**6**) RVA vTLP (VP2/6/7); (**7**) RVA vTLP (VP2/6/4/7); (**8**) RVC vTLP (VP2/6/7). (**B**) VP6 thin helical 45 nm thick tubes (top) and thick 75 nm tubes (bottom). Scale bar 100 nm. Adapted from [134,142,147,155].

VLPs maintain the morphology and antigenic characteristics of native viruses while lacking genetic material, making them non-infectious. They exhibit higher immunogenicity compared to monomeric recombinant proteins [133]. VLP-based vaccines can induce innate and adaptive immune responses through various immunization routes [156,157,158,159] and potentially reduce the side effects associated with conventional vaccines [160,161,162]. As a result, VLPs are promising candidates for the development of vaccines [162], particularly for emerging RVA serotypes [156,157,158,159] or in combination with other enteric pathogens [163]. Other applications of these VLPs are related to the characterization of neutralizing and non-neutralizing epitopes in VP4 and VP7 by ELISA and hemagglutination assays [134,164]. The VLPs play a key role in VLP-based ELISA for the quantitation of RV antibodies, especially for strains that do not replicate in in vitro cultures.

VLPs are also of interest in the medical and biotechnology fields as drug delivery agents (nanocarriers) [165,166,167] or vehicles to display different heterologous epitopes [168]. The expression of a foreign protein at the amino terminus of VP2 does not prevent VLP formation and could protect the molecules contained within it from degradation and, in some cases, also enhance its uptake [169]. VP6 assemblies can be used to produce hybrid nanobiomaterials [170] by functioning as a multimeric scaffold for the in situ synthesis of noble metal, magnetic, and semiconductor nanoparticles conjugated over the reactive amino acid residues of VP6. These hybrid nanobiomaterials exhibit high morphological consistency, with potential applications in material sciences and nanomedicine.

## 7. Future Outlook

As we have observed, research on rotavirus is thriving, as evidenced by the abundance of relevant results within the scope of this review. This progress is in large part driven by recent technological advancements, particularly the development of a reverse genetics (RG) system for rotaviruses [130,171,172,173] and the revolutions in various microscopy techniques that we are currently witnessing [174]. As we have seen in this review, these advances have had a significant impact on rotavirus research, providing important insights into viral replication, assembly, and cell entry mechanisms. Furthermore, these improvements have enabled the use of rotaviruses as vectors for the expression of proteins and peptides and have deepened our understanding of rotavirus pathogenesis [58,175,176,177,178,179]. The fully plasmid-based RG system, which provides complete control over the genome structure, has emerged as the preferred platform for the development of next-generation rotavirus vaccines [180,181,182,183,184].

These critical experiments have not only answered numerous questions about rotavirus biology but also stimulated new areas of enquiry:The current molecular model for membrane disruption and entry is being developed in an outlier system (RRV strain/BSC-1 cells) that does not depend on the cellular endocytic machinery and instead relies solely on the viral machinery for entry from the plasma membrane. How does this model apply to other rotavirus strains that do rely on the cellular endocytic machinery and enter the cell from late or recycling endosomes?What are the mechanisms underlying the entry and non-lytic release of rotavirus as vesicle-enveloped clusters, and how does vesicle-mediated en bloc transmission affect viral pathogenesis, spread, and evolution?Identifying the membrane in eDLP as of COPII origin is a paradigm-shifting result that awaits further characterization of the mechanisms involved. Additionally, it is crucial to investigate the extent to which rotavirus disrupts the autophagy system and the specific mechanisms underlying VP7 transport within these membranes.The description of viroplasms as biomolecular condensates is crucial in understanding the dynamics of these viral organelles. To develop a comprehensive understanding of the formation and maturation of viroplasms, it is important to address key questions related to the maturation of viroplasms, the roles of lipid droplets and other cellular components in the formation of viroplasms, and the roles of NSP2 and VP2 in the formation and evolution of initial genome complexes.The description of viroplasms as highly organized structures with a concentric distribution of viral proteins around a center formed by NSP5 suggests that there exists a temporal and spatial coordination of viral processes, which has yet to be characterized in detail.Cryo-electron tomography of cellular lamellae and subtomogram averaging have provided valuable insights into the structure of eDLP and have been demonstrated as a promising approach to characterizing transient rotavirus structures in their native cellular environment. With further advancements in these techniques, it is anticipated that the reconstruction of VP7 and NSP4 within the eDLP can be achieved.The presence and the structure of the genome-less SLP are difficult to conciliate with current models for rotavirus particle assembly and require further investigation.

## Figures and Tables

**Figure 1 viruses-15-01750-f001:**
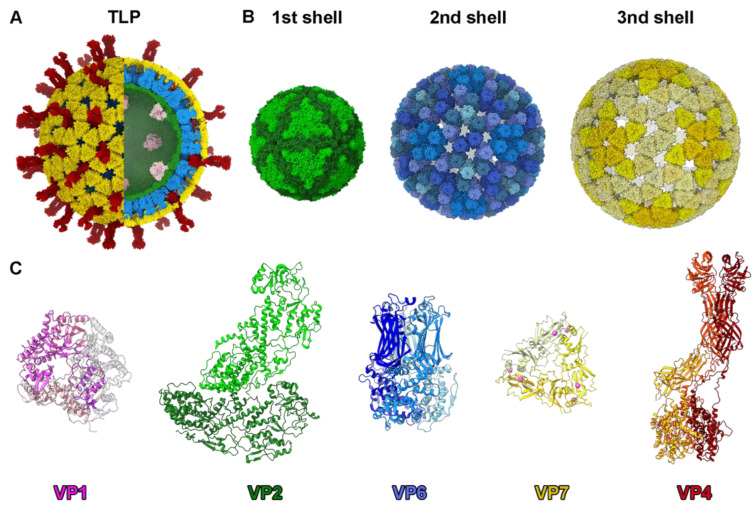
Rotavirus particle structure. (**A**) Overview of the TLP structure in a hybrid model built from the atomic model of the RdRp VP1 attached to the inner core (PDB 3OJR [21]) and the TLP cryoEM structure (PDB 4V7Q [20]). Color code is indicated in panel C. (**B**) Structure of the three concentric icosahedral layers. (Left) VP2 T = 1 inner shell. The two types of VP2 conformers (VP2-A and VP2-B) are indicated and colored with different levels of green. (Center) Structure of the intermediate VP6 T = 13 capsid. The 13 VP6 monomers of the asymmetric unit are arranged into 5 types of trimeric capsomers colored with different levels of blue. (Right) Structure of the outer layer. The VP7 glycoprotein trimers are localized in phase with the VP6 trimers in a T = 13 architecture. The five types of trimeric capsomers are colored with different levels of yellow. (**C**) Atomic structure of RV structural proteins. From left: VP1 RdRp (pink), VP2 dimer (VP2A dark green, VP2B light green), VP6 trimer (monomers in different blue levels), VP7 trimer (monomers in different levels of yellow and Ca^2+^ ions in magenta), and VP4 trimer (subunits highlighted in different levels of red) [20,22,23].

**Figure 4 viruses-15-01750-f004:**
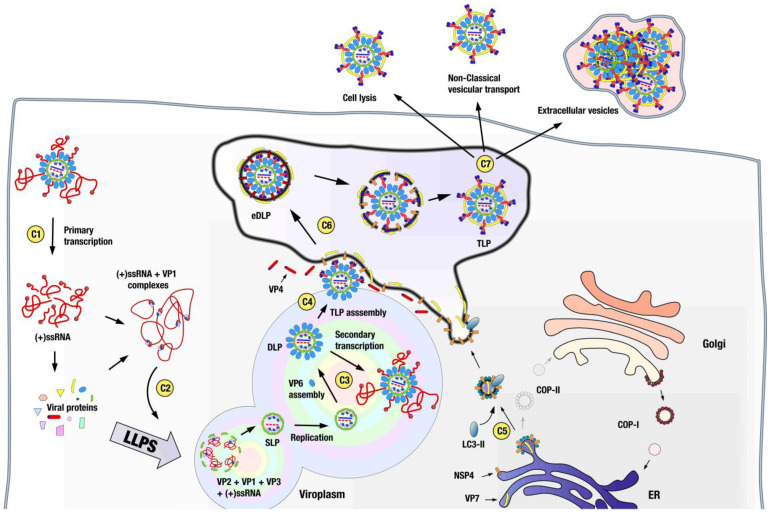
Assembly of virions. Primary transcription occurs immediately after the release of the DLP to the cytoplasm, and capped, non-polyadenylated, positive-sense ssRNAs are extruded from the viral particle, initiating the synthesis of viral proteins (C1). Assembly begins with the specific interaction of proteins VP1 (and possibly VP3) with conserved bases at the 3′ ends of the ssRNA(+), forming pre-core complexes. The 11 different complexes are specifically recruited to the viroplasms (C2), which are biomolecular condensates formed by the liquid–liquid phase separation (LLPS) of viral proteins NSP5 and NSP2, and where the generation of SLP, genome replication, DLP assembly, and secondary transcription occur (C3). TLP assembly occurs within membranous structures surrounding viroplasms, which are derived from COOP-II cisternae (C4), which are sequestered and diverted, along with viral protein VP7, to the vicinity of the viroplasms by the non-structural protein NSP4 (C5). Interaction between DLP-VP6 and NSP4 drives the progressive engulfment of the DLP by NSP4/VP7-containing membranes, resulting in the budding of the DLP–VP4–NSP4–VP7 complex into the lumen of the COPII-derived vesicles, in the form of transient membrane-enveloped particles (eDLP) (C6). The rupture of the eDLP envelope appears to be directed by conformational changes in the spike protein VP4, which transitions from a highly flexible, premature conformation in the eDLP to the partially dimeric mature conformation observed in the TLP. The disruption of the transient envelope and the assembly of the VP7 capsid, which locks the VP4 spikes in place, are proposed to be driven by this transition. The process by which the newly assembled TLP, within COPII-derived vesicles, moves outside the cell has not been thoroughly investigated (C7). However, in different systems, rotavirus has been shown to exit the cell by lysis, active secretion from the apical cell surface before cell lysis occurs, and in the form of extracellular vesicles containing viruses originating at the plasma membrane, which are responsible for the vesicle-mediated en bloc transmission.

**Figure 5 viruses-15-01750-f005:**
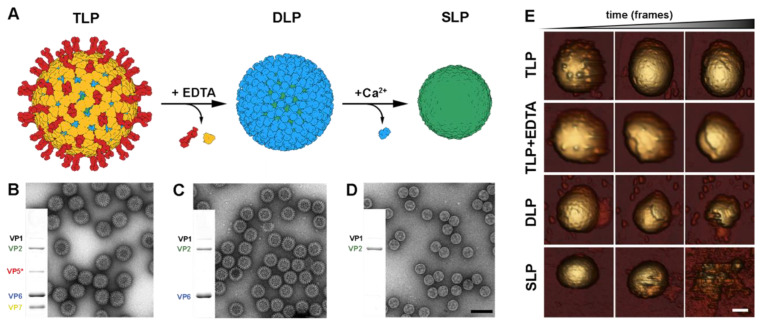
In vitro disassembly of RV particles. (**A**) Generation of DLP and SLP from TLP. In the presence of EDTA, VP7 and VP5*/VP8* are disassembled from TLP. The liberation of SLP is achieved through the dissociation of VP6 trimers upon exposure to a high concentration of Ca^2+^ ions. (**B**–**D**) Characterization of TLP, DLP, and SLP using Coomassie-blue-stained SDS-PAGE gels and negative staining electron microscopy. The gel images show the positions of rotavirus structural proteins (VP). The scale bar represents 100 nm. (**E**) Topographic evolution of TLP, TLP + EDTA, DLP, and SLP during continuous AFM imaging at low force. Sale bar represents 20 nm [29].

**Figure 6 viruses-15-01750-f006:**
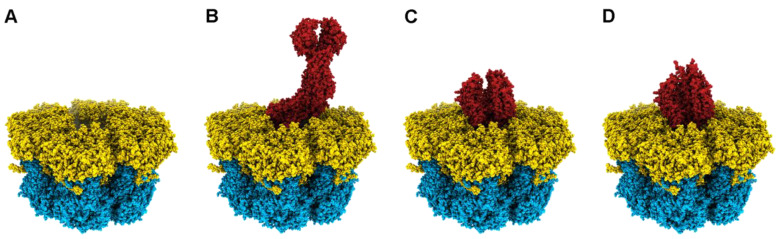
Cryo-EM structures of recoated DLPs. Hexameric position of a DLP section recoated with (**A**) VP7 (PDBs 3GZT and 3GZU). (**B**) VP7 + VP5*/VP8* in upright conformation (PDB 6WXE). (**C**) VP7 + VP5*/VP8* in intermediate conformation (PDB (6WXF). (**D**) VP7 + VP5*/VP8* in reversed conformation (PDB 6WXF) [49].

## Data Availability

Not applicable.

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
