# Peer review of "Rotavirus Particle Disassembly and Assembly In Vivo and In Vitro"

_viruses, 2023, doi:10.3390/v15081750_

Round 1

Reviewer 1 Report

Rotavirus particle assembly and disassembly

By D Asensio-Cob et al (Corresponding authors: J M Rodriguez, D Luque)

Submitted to Viruses (Editorial No. viruses-2452421)

General Comments

This is a review on classical and recent studies aimed at elucidating the structural and functional features of rotavirus (RV) and cellular proteins during the viral replication cycle at a molecular level. Rotaviruses are triple-layered particles enclosing the genome of 11 segments of dsRNA and two proteins involved in transcription/replication functions. Each particle layer contains particular viral proteins with different functions. Most of the relevant publications have been mentioned and assessed in context. The review concentrates on events occurring during disassembly of RV particles in the course of the cellular entry steps and during assembly/early viral morphogenesis. Major areas of progress include: cryo-electron tomography, recognition of viroplasms as biomolecular condensates, and the recent availability (5 years) of plasmid only-based reverse genetics (RG) systems, and their significance for progress in RV research has been fully recognized.

While most relevant facts and data are appropriately considered, the citation of references contains duplications and triplications to a large extent. Thus, the reference list will require a complete and extensive overhaul and renumbering (with attention to concordance of numbers in reference list and text). Numerous minor clarifications are requested, and some additional recent references are suggested.

Specific Comments

Line

2          Reconsider title, e.g. ‘Rotavirus particle disassembly and assembly in vivo and in vitro’. Since the text flow is orientated along the steps of the viral replication cycle, the order of ‘disassembly and assembly’ appears to be preferable and should be considered for various places in the text as appropriate (see comment below8. Since the text also contains data on the production of various virus-like particles (VLPs), this should be apparent from the title.

8          Read: … Electron microscopy unit…

19f       See comment line 2. (also in line 42, 65, 67, 377 etc)

24        Consider reading: … During … cycle, double-stranded RNA (dsRNA) viruses face…

34        … without disassembling the core particle…

43        … rotaviruses (RVs), together with… virus, are the model system…

49        … vaccines has led to their inclusion in…

50        The RV-associated death of children <5 y of age was considered to be 128,500 worldwide in 2016 (Troeger et al, 2018, ref 15). Also consider: GBD 2016 Diarrhoeal Disease Collaborators. Estimates of the global, regional, and national morbidity, mortality, and aetiologies of diarrhoea in 195 countries: a systematic analysis for the Global Burden of Disease Study 2016. Lancet Infect Dis. 2018 Nov;18(11):1211-1228.

66        … and its subviral particles, and review studies supporting…

72        … enclosing eleven…

74        … pentameric symmetry positions…

76        Consider adding: Mathieu M, Petitpas I, Navaza J, Lepault J, Kohli E, Pothier P, Prasad BV, Cohen J, Rey FA. Atomic structure of the major capsid protein of rotavirus: implications for the architecture of the virion. EMBO J. 2001 Apr 2;20(7):1485-97.

78f       … [19, 25, 26]… that account for …

88        Ref. 28 duplicates ref. 22.

100      Provide a ref for the tetrameric structure of VP3.

106      Ref. 31 duplicates ref. 20. Ref. 32 duplicates ref. 21.

107      After. Fig. 1. Provide a reference for the 5 different structures of VP6 capsomers (refs 23, 24?)

119      … polymerase activity [21, 24]. …

129      Ref. 37 duplicates ref. 32.

133      Ref. 38 duplicates ref. 26.

134      … the interaction between both proteins being primarily facilitated by… .

154      … The beta-barrel of one subunit forms the spike stalk…

164      Print Reoviridae in italics.

166      After. Fig. 2 Legend. Spell out the abbreviations of the different enzymatic functions of the VP3 subdomains. Provide a reference. Cite, describe and assess: Kumar D, Yu X, Crawford SE, Moreno R, Jakana J, Sankaran B, Anish R, Kaundal S, Hu L, Estes MK, Wang Z, Prasad BVV. 2.7 Å cryo-EM structure of rotavirus core protein VP3, a unique capping machine with a helicase activity. Sci Adv. 2020 Apr 15;6(16):eaay6410.

The color code of the 2’O-MTase domain should read: yellow.

167      Omit ‘with an extracellular phase’.

169      Ref 45 duplicates ref 1.

170      … produce ss(+)RNAs that function as mRNA… or as precursors of…

175      Ref. 48 duplicates ref 33.

182      Ref. 56 duplicates ref 40. Ref. 57 duplicates ref 42.

187      Ref. 59 duplicates ref 19.

212      Read: … [67, 71, 72, 74].

224      Fig. 3, Legend. Explain steps B1-B5. Line 2, read: … which release the lectin domains of… from the virus… Line 3, read: … endocytosis, depending on strain, could be… Penultimate line: a reference should be given for vesicle-cloaked en bloc transmission.   Consider citation and review of: Santiana M, Ghosh S, Ho BA, Rajasekaran V, Du WL, Mutsafi Y, De Jésus-Diaz DA, Sosnovtsev SV, Levenson EA, Parra GI, Takvorian PM, Cali A, Bleck C, Vlasova AN, Saif LJ, Patton JT, Lopalco P, Corcelli A, Green KY, Altan-Bonnet N. Vesicle-Cloaked Virus Clusters Are Optimal Units for Inter-organismal Viral Transmission. Cell Host Microbe. 2018 Aug 8;24(2):208-220.e8.

243      Read: … perforation and DLP release into the cytoplasm…

250      … the tight-fitting membranes… [Correct?]

257      Read: … precedes the onset of… dissociation by approximately 2 min…

266      … released into the cytoplasm…

271      Extensive change of numbered refs required.

273      … It is in the viroplasms where…

274      … The generation of… and secondary transcription occur in …

279      … condensates [83, 98-100] …

290      Particle assembly progresses…

291      Regarding packaging of RV RNA consider citation of: Moreno-Contreras J, Sánchez-Tacuba L, Arias CF, López S. Mature Rotavirus Particles Contain Equivalent Amounts of 7meGpppG-Capped and Noncapped Viral Positive-Sense RNAs. J Virol. 2022 Sep 14;96(17):e0115122.

Regarding the RNA packaging capacity of RVs and other genera of the Reoviridae consider: Desselberger U. What are the limits of the packaging capacity for genomic RNA in the cores of rotaviruses and of other members of the Reoviridae? Virus Res. 2020 Jan 15;276:197822.

300      … 5-fold symmetry axes due to displacement of…

310      … [78, 20, 29, 107] …

323      … progresses by…

339      … by 2 loops per molecule…

363      After. Fig. 4. In the viroplasm component at the lower left, the LLPS data are missing; they should at least be mentioned in the legend. Preferably, the figure should be modified.

            Legend line 5. … auto-inhibited… Please clarify.                                                                           Line 6: … large structures of viral origin and where generation… and secondary transcription occur.

375      … are infectious…

388      Ref 138 is a duplicate of ref 67.

410      After. Fig. 5, Legend. The reference [137] from which these data are taken should be indicated.

416      Read: … than native virions …

423      … and a lower effect at pH 7.2.

431      Read: … [152, 155].

432      … but for full restitution of infectivity, VP4 must be added before VP7.

435ff   Reverse genetics of rotaviruses. Two key refs should be added:

Kanai Y, Komoto S, Kawagishi T, Nouda R, Nagasawa N, Onishi M, Matsuura Y, Taniguchi K, Kobayashi T. Entirely plasmid-based reverse genetics system for rotaviruses. Proc Natl Acad Sci U S A. 2017 Feb 28;114(9):2349-2354.

Komoto S, Fukuda S, Ide T, Ito N, Sugiyama M, Yoshikawa T, Murata T, Taniguchi K. Generation of Recombinant Rotaviruses Expressing Fluorescent Proteins by Using an Optimized Reverse Genetics System. J Virol. 2018 Jun 13;92(13):e00588-18.

In addition a short paragraph should be devoted to the use/application of RVA RG as a vector for the expression of heterologous peptides/proteins and the use of recombinant RVs for pathogenesis studies. The following should be considered for citation:

Philip AA, Patton JT. Expression of Separate Heterologous Proteins from the Rotavirus NSP3 Genome Segment Using a Translational 2A Stop-Restart Element. J Virol. 2020 Aug 31;94(18):e00959-20.

Philip AA, Patton JT. Rotavirus as an Expression Platform of Domains of the SARS-CoV-2 Spike Protein. Vaccines (Basel). 2021 May 3;9(5):449. doi: 10.3390/vaccines9050449. PMID: 34063562; PMCID: PMC8147602.

Philip AA, Patton JT. Rotavirus as an Expression Platform of Domains of the SARS-CoV-2 Spike Protein. Vaccines (Basel). 2021 May 3;9(5):449. doi: 10.3390/vaccines9050449. PMID: 34063562; PMCID: PMC8147602.

Kawagishi T, Sánchez-Tacuba L, Feng N, Costantini VP, Tan M, Jiang X, Green KY, Vinjé J, Ding S, Greenberg HB. Mucosal and systemic neutralizing antibodies to norovirus induced in infant mice orally inoculated with recombinant rotaviruses. Proc Natl Acad Sci U S A. 2023 Feb 28;120(9):e2214421120.

Zhu Y, Sánchez-Tacuba L, Hou G, Kawagishi T, Feng N, Greenberg HB, Ding S. A recombinant murine-like rotavirus with Nano-Luciferase expression reveals tissue tropism, replication dynamics, and virus transmission. Front Immunol. 2022 Jul 29;13:911024.

Sánchez-Tacuba L, Kawagishi T, Feng N, Jiang B, Ding S, Greenberg HB. The Role of the VP4 Attachment Protein in Rotavirus Host Range Restriction in an In Vivo Suckling Mouse Model. J Virol. 2022 Aug 10;96(15):e0055022.  [ref 66]

Yamasaki M, Kanai Y, Wakamura Y, Kotaki T, Minami S, Nouda R, Nurdin JA, Kobayashi T. Characterization of Sialic Acid-Independent Simian Rotavirus Mutants in Viral Infection and Pathogenesis. J Virol. 2023 Jan 31;97(1):e0139722.

In addition, the usefulness of RV RG for the development of next generation RV vaccines may be noted. E.g.

Desselberger U. Potential of plasmid only based reverse genetics of rotavirus for the development of next-generation vaccines. Curr Opin Virol. 2020 Oct;44:1-6.

Kanai Y, Onishi M, Kawagishi T, Pannacha P, Nurdin JA, Nouda R, Yamasaki M, Lusiany T, Khamrin P, Okitsu S, Hayakawa S, Ebina H, Ushijima H, Kobayashi T. Reverse Genetics Approach for Developing Rotavirus Vaccine Candidates Carrying VP4 and VP7 Genes Cloned from Clinical Isolates of Human Rotavirus. J Virol. 2020 Dec 22;95(2):e01374-20.

Kanai Y, Kobayashi T. Rotavirus reverse genetics systems: Development and application. Virus Res. 2021 Apr 2;295:198296.

Kanai Y, Kobayashi T. FAST Proteins: Development and Use of Reverse Genetics Systems for Reoviridae Viruses. Annu Rev Virol. 2021 Sep 29;8(1):515-536.

Kobayashi T, Patton JT, Desselberger U. Species A rotavirus reverse genetics: Achievements and prospects. Virus Res. 2021 Dec;306:198583.

440      … which are at present very difficult to generate…

443      Ref. 159 is a duplication of ref 55.

            After.Fig. 6, legend. … Hexameric position of a DLP section recoated with…

444      Sections 5 and 6 of the ms require a revision for more precise separation of data obtained by in vivo and in vitro approaches.

454      Please provide refs for this statement.

466f     … Fig. 7A, 6… Fig. 7A, 5…

470      Explain RVA and RVC at first mentioning.

497      For sections 4-6 of the manuscript it may be useful to screen the following review:

Desselberger U. 14th International dsRNA Virus Symposium, Banff, Alberta, Canada, 10-14 October 2022. Virus Res. 2023 Jan 15;324:199032.

518ff   References

            This section will require a complete overhaul, due to various duplications and some triplications, e.g. of refs 1, 19, 20, 21, 22, 26, 29, 36, 40, 42, 55, 67, 152. [There may be more.]

            The following refs are incomplete: 3, 5, 8, 10, 11, 12, 47, 132, 149, 160.

            The following refs are not cited in the text:  142-144, 150, 161-172, 196-198, 214-216.

            Ref. 5. Consider citation of the 2013 edition of Fields Virology.

            Check correctness of refs 30, 149.

Moderate editing of English is required.

Reviewer 2 Report

Rotaviruses are 11-segmented, double-stranded RNA (dsRNA) viruses and important causes of gastroenteritis in young animals, including human infants. The rotavirus virion is a non-enveloped, triple-layered particle, which undergoes orchestrated disassembly and assembly processes during host cell infection. Here, Asensio-Cob et al. reviews current models of how such disassembly/assembly might occur, with an emphasis on the structures and functions of the viral proteins involved.

In general, this article well written and accurate; it will be of great value to the rotavirus field. However, some comments related to early particle assembly are overstated and not fully consistent with the available data. The authors should consider tempering their language in this regard, and perhaps throughout the document, to emphasize which ideas are strongly rooted in primary data versus which ideas are hypotheses/models with little supportive data. It would also be helpful if the authors ended the article with a section that highlights the major gaps in knowledge/areas of uncertainty related to rotavirus disassembly/assembly so as to stimulate future work.

Several additional editorial suggestions are also listed below for the authors’ consideration.

1.     Line 14 and throughout: I don’t think that there is any strong evidence that selective packaging (or replication of the of the viral genome for that matter) occurs in the context of single-layered particles (SLPs). Selective assortment could occur prior to packaging. Replication-active assembly intermediates have been shown to contain VP6, and thus, might not be SLPs-see also comment 6 below. Please consider rephrasing.

2.     Line 37 and throughout: Please use the updated ICTV nomenclature (PMID: 36215107). Rotavirus belongs to the new order Reovirales and the Sedoreoviridae family.

3.     Line 50: Please add a reference for the claim that rotavirus continues to cause 230,000 deaths per year. My understanding is that current numbers are much lower (<130,000). Perhaps there is a more up to date reference than Troeger’s 2017, 2018 papers.

4.     Lines 93-98: The VP2 N-termini are very flexible, and as such, they are difficult to resolve using structural biology techniques. The DLP structures published by Jenni and Ding show only a portion of the N-terminal extensions (residues ~70-134), with 3 of them contacting VP1. That doesn’t mean that only 3 make contacts; it is just what can be seen. Consider rephrasing and adding this nuance. Indeed, the vast majority of VP2 N-terminal density around the fivefold (residues ~1-70 from 10 monomers=>700 amino acids) are not observed in the structure shown in Fig. 2A.

5.     Line 119: Please change “this asymmetry has been linked to regulation of polymerase activity” to “ this asymmetry has been proposed to regulate the polymerase”. To my knowledge, there is no biochemical evidence in the reference paper regarding polymerase activity.

6.     Lines 308-312: There is insufficient evidence about whether VP2 SLPs (i.e., lacking VP6) are formed inside of the infected cells. The genome-less SLPs reported by Shah et al, 2023 might just be abortive assembly by-products. While VP6 is not required for genome replication, replicase-active early assembly intermediates purified from infected cells do, indeed, contain VP6...and they don't look like SLPs. The authors should consider revising and citing the following references: PMID: 2845649 and PMID: 25635339.

7.     Line 319 and throughout: Please change “COOPII” be “COPII”

8.     Line 416 is not English. Please translate.

9.     Lines 456-457: Recombinant VP2 is highly insoluble and forms a mix of multimers, including bristly structures, not just SLPs.  Please see: PMID: 8178489.

10.  Line 506: The review article ends abruptly. A final section highlighting key gaps in knowledge/gaps in technologies to which future research should focus on would be very valuable.

11.  Fig. 1 legend: Change “RdRd” to “RdRp” and (D) to (B). Both VP2 monomers appear in dark green.

12.  Fig. 3. The authors should consider adding a title to this legend.

13.  Fig. 4. Is there evidence that VP1/VP3/+ssRNA complexes form in the cytosol? Please cite. Recent work by Strauss et al. suggests that viroplasmic-localized VP1 is required for +RNA recruitment (PMID: 36700549), perhaps challenging this older model.

14.  Fig. 5. Are the EM and AFM images from a primary paper? If so, please add reference.

15.  Fig. 7. It is not clear where these EM images are derived from; references are book chapters. Please add primary paper figure references. Also, the scale of the VP2 image looks very different than the other images. As mentioned above, recombinant VP2 forms lots of different structures as a recombinant protein, not just SLPs (see comment 9). 

16.  References: Many references are repeated multiple times. See 8 and 49; 20, 39 and 71; 21 and 32 for examples. Please carefully cross-check and edit.

This is a nice review article, and I suggest minor revisions prior to publication.

Round 2

Reviewer 1 Report

Rotavirus particle disassembly and assembly in vivo and in vitro

By Dunia Asensio-Cob et al

(Corresponding authors: Javier M Rodriguez, Daniel Luque)

Submitted to Viruses (Editorial No. viruses – 2452421 R1)

General Comments

This is the revised version of a manuscript the original version of which has been studied and commented upon by this reviewer. The authors have accepted the vast majority of comments/suggestions, and the manuscript has significantly improved. However, this reviewer has a number of additional comments on the revised version (see Specific Comments), which should be considered for further improvement of the manuscript. In particular, the ref. list and the numbering of refs in text and list require continuing attention.

Specific Comments

Line (according to revised version with changes in red)

15        Consider reading: Rotaviruses (RVs) are … of human infants and of the young of a large number of animal species. The viral particle…

27        … and cellular RV biology that have…

39        …host responses [2, 3]. Most…

54        … Due to their clinical relevance…

61        … in >100 countries…

62        … confidence interval…

66        Ref 17 is published in 1974 and thus not relevant in the context. Its citation in line 74 is appropriate.

75        … various structural… studies have revealed…

77        … are disassembled and assembled…

83        … that surround…

88        … is surrounded by…

104      A sentence should be added describing Fig. 1C.

122      After. Fig. 1 legend, line 3. Read: … in panel C. (B) Structure of…

152      … that calcium ions do not only stabilize… but also serve…

226      … due to an unknown mechanism, by the nature…

245      After. Fig. 3, legend. Line 8.  …late endosome compartment (B7) before … Last line: …but remains to be characterized further.

250      … domains, adopting a trimeric structure…

275      … beta-barrel, implying that…

304      … condensates [75, 76, 90, 91].  Ref. 92 is a duplication of ref 75. All subsequent refs will have to be renumbered (in text and ref list). See also comment below.

307      … solid…

309      … phosphorylation…

370      … by 2 loops to their VP4 molecule

396      After. The expansion of Fig. 4 is appropriate in its diagrammatic form.

471      … VP4 must be added before VP7 [36]. VP4 spikes…

489      Ref. 148. There is a gap in refs numbering due to the fact that refs 134-147 have been moved to ‘Future outlook’ (lines 551ff). Renumbering of refs [134ff] in the order of first citation is required.

516      … VLP, such as yeast…

554      See comment line 489.

563      … have not already answered numerous… but will also enable new areas of enquiry: …

567f     … relies… that do rely on…

593      … for rotavirus particle assembly…

607ff   References

            The following refs are incomplete: 5, 11, 12, 52, 95, 96, 98, 106, 122, 133, 148, 150, 156, 186.

            Ref. 9. Clarify.

            Ref.92 is a duplication of ref. 75.

            Refs 134 ff require renumbering. See comment above.

Moderate editing of English is required.
